# The High Proportion of Discordant *EGFR* Mutations among Multiple Lung Tumors

**DOI:** 10.3390/cancers14123011

**Published:** 2022-06-18

**Authors:** Hyunwoo Lee, Jin Hee Park, Joungho Han, Young Mog Shim, Jhingook Kim, Yong Soo Choi, Hong Kwan Kim, Jong Ho Cho, Yoon-La Choi, Wan-Seop Kim

**Affiliations:** 1Department of Pathology and Translational Genomics, Samsung Medical Center, Sungkyunkwan University School of Medicine, Seoul 06351, Korea; hwpatho.lee@samsung.com (H.L.); hanjho@skku.edu (J.H.); 2Department of Pathology, Konkuk University Medical Center, Konkuk University School of Medicine, Seoul 05030, Korea; 20170403@kuh.ac.kr; 3Department of Thoracic and Cardiovascular Surgery, Samsung Medical Center, Sungkyunkwan University School of Medicine, Seoul 06351, Korea; youngmog.shim@samsung.com (Y.M.S.); jhingook.kim@samsung.com (J.K.); ysooyah.choi@samsung.com (Y.S.C.); hkts.kim@samsung.com (H.K.K.); jongho9595.cho@samsung.com (J.H.C.); 4Department of Health Science and Technology, Samsung Advanced Institute for Health Sciences and Technology, Sungkyunkwan University, Seoul 06351, Korea

**Keywords:** lung cancer, non-small cell lung carcinoma, multiple pulmonary nodules, epidermal growth factor receptor

## Abstract

**Simple Summary:**

Lung cancer is one of the leading causes of cancer-related mortality worldwide. The incidence of multiple primary lung cancers has been increasing. In addition to the identification of epidermal growth factor receptor (EGFR) tyrosine kinase inhibitors, the evaluation of the *EGFR* mutation status in lung cancer is important to devise optimal treatment strategies. In this study, the *EGFR* mutation status in multiple primary lung cancers was examined, and its discordance rate in individual tumors was determined to be high. Our findings reveal the importance of *EGFR* mutation analysis in individual tumors of multiple primary lung cancers.

**Abstract:**

The prevalence of multiple lung cancers has been increasing recently. Molecular analysis of epidermal growth factor receptor (*EGFR*) mutations in individual tumors of multiple lung cancers is essential for devising an optimal therapeutic strategy. The *EGFR* mutation status in multiple lung cancers was evaluated to determine its therapeutic implications. In total, 208 tumors from 101 patients who underwent surgery for multiple lung cancers were analyzed. Individual tumors were subjected to histological evaluation and *EGFR* analysis using a real-time polymerase chain reaction. Additionally, *EGFR*-wildtype tumors were subjected to next-generation sequencing (NGS). *EGFR* mutations were detected in 113 tumors from 72 patients, predominantly in females (*p* < 0.001) and non-smokers (*p* < 0.001). Among patients with at least one *EGFR*-mutant tumor, approximately 72% of patients (52/72) had different *EGFR* mutations in individual tumors. NGS analysis of *EGFR*-wildtype tumors from 12 patients revealed four and eight cases with concordant and discordant molecular alterations, respectively. These findings revealed a high proportion of discordant *EGFR* mutations among multiple lung tumors. Hence, *EGFR* analysis of individual tumors of multiple lung tumors is essential for the evaluation of clonality and the development of an optimal treatment strategy.

## 1. Introduction

Lung cancer is one of the leading causes of mortality worldwide [1]. Primary lung cancer, which is associated with multifocal presentations, accounts for 0.2–8.1% of all lung cancer cases [2,3,4,5]. The incidence of multiple lung cancers, which are classified as synchronous (simultaneous detection of cancerous lesions) and metachronous (a new cancer lesion is detected six months post-first-cancer lesion detection), and multiple primary lung cancers, has markedly increased. The pathogenesis of adenocarcinoma has been widely examined. However, the mechanisms underlying the pathogenesis of multiple adenocarcinomas have not been elucidated. The multifocal presentations of squamous cell carcinoma, which is closely linked to smoking, can be explained by ‘field cancerization’ [6]. However, the mechanism of adenocarcinoma development, which involves a multi-step process from atypical adenomatous hyperplasia (AAH) to the occurrence of invasive cancerous lesions [7], cannot efficiently explain the mechanism of multifocal presentations.

Several studies have suggested the potential mechanisms underlying the development of multiple adenocarcinomas. The detection of *EGFR* mutations in the non-neoplastic epithelium around the adenocarcinoma indicates that *EGFR* mutations occur in the early stages of carcinogenesis [8]. Alternatively, independently occurring *EGFR* mutations play an important role in multifocal carcinogenesis as evidenced by cancerous lesions, such as microinvasive adenocarcinoma (MIA) or invasive adenocarcinoma exhibiting an increased frequency of *EGFR* mutations when compared with AAH lesions [9]. Irrespective of the timing of the occurrence of mutations, analysis of *EGFR* mutations has been a crucial step for the application of tyrosine-kinase-inhibitor (TKI)-based targeted therapy in patients with lung cancer harboring *EGFR* mutations [10].

Distinguishing multifocal primary lung cancers (MPLC) from intrapulmonary metastasis (IPM) has been a long-standing clinical challenge in multiple lung cancers. Traditionally, MPLC and IPM are distinguished based on the clinical stage and histological features [11,12,13,14]. However, the histological characteristics of the primary tumor may be discordant with those of metastatic foci. Recent studies have suggested that the genetic similarity between a primary tumor and metastatic foci must be determined using next-generation sequencing (NGS) [15,16,17]. However, performing NGS on all tumors is not feasible in general clinical settings. In cases of multiple cancers, only one typical cancerous lesion is subjected to *EGFR* analysis. Thus, the response of the tumors to EGFR-TKI treatment may be different from the expected response.

Adenocarcinoma accounts for 40.3–91.3% of all multiple primary lung cancer cases [5,18,19]. The frequency of *EGFR* mutation occurrence in adenocarcinomas is 12–48% [20,21]. Therefore, the actual prevalence of *EGFR* mutation in MPLC can be higher than the expected prevalence. Recently, the US Food and Drug Administration approved osimertinib, a third-generation TKI, as an adjuvant treatment for early-stage, non-small cell lung cancer (NSCLC). To establish appropriate treatment guidelines for MPLC, the prevalence of *EGFR* mutation and the potential for discordant *EGFR* status in individual tumors of MPLC must be evaluated. This study aimed to examine the diverse *EGFR* mutation statuses in multifocal cancer lesions. Based on the *EGFR* status, the optimal treatment for patients can be determined. Additionally, the applicability of the criteria used to distinguish tumors from metastases was confirmed.

## 2. Materials and Methods

### 2.1. Case Selection and Clinicopathologic Review

All patients who underwent surgery for multiple lung cancers between 2017 and 2020 at the Samsung Medical Center (Seoul, Korea) were enrolled in this study. The results of patients with multiple lung cancers who underwent *EGFR* analysis for at least two tumors were reviewed. All histopathological findings of hematoxylin-and-eosin (H&E) stained sections were evaluated by two pathologists (Y.-L.C. and H.L.). Cases with obvious IPM, as determined based on both histological and clinical assessments, were excluded. The cases were classified as MPLC or IPM based on the standard Martini–Melamed criteria (Appendix A) for multiple lung cancers [11], definitions and guidelines for MPLC proposed by the American College of Chest Physicians (ACCP) (Appendix A) [12,13], and the comprehensive histological assessment proposed by Girard et al. (Appendix A) [14].

This study was approved by the Institutional Review Board (IRB) of Samsung Medical Center (IRB No. 2020-04-071-001, 2022-01-009-001). The written informed consent was waived due to anonymous data analysis.

### 2.2. EGFR Analysis

*EGFR* mutation analysis was performed using the cobas^®^ EGFR mutation test v2 (Roche Molecular Systems, Inc., Pleasanton, CA, USA). A list of detectable *EGFR* mutations is summarized in Appendix A. Formalin-fixed paraffin-embedded (FFPE) tissues were sectioned into a thickness of 5 μm. The sections were deparaffinized and subjected to genomic DNA extraction using a cobas^®^ DNA sample preparation kit (Roche Molecular Systems, Inc.). The isolated genomic DNA was quantified using spectrophotometry, following the manufacturer’s instructions. Next, the genomic DNA (150 ng) was amplified and detected using a cobas^®^ z480 analyzer (Roche Molecular Systems, Inc.). All results were automatically described and reported as positive, negative, or invalid using the cobas^®^ 4800 software (Roche Molecular Systems, Inc.).

### 2.3. Sample Preparation and DNA Extraction for NGS

After the histological assessment of H&E-stained sections by a pathologist to confirm tumor cell contents (tumor purity), the tumor areas of the FFPE sections were macro-dissected. The tumor cellularity for NGS was set at 10%. The FFPE samples were sectioned into 4 μm thick sections, and 5–10 slides of unstained tissue were prepared. The sample was deparaffinized using xylene and 100% ethanol. Genomic DNA (20 ng per sample) and RNA (20 ng per sample) were extracted using a RecoverAll total nucleic acid isolation kit (Thermo Fisher Scientific, Waltham, MA, USA), following the manufacturer’s instructions. DNA and RNA concentrations were examined using a Qubit 3.0 Fluorometer (Thermo Fisher Scientific) with a Qubit dsDNA HS assay kit and a Qubit RNA HS assay kit (Thermo Fisher Scientific), respectively.

### 2.4. Library Preparation and Sequencing

The Oncomine comprehensive assay v1 (Thermo Fisher Scientific), which examines 143 genes, was performed to detect single nucleotide variants (SNVs), copy number alterations (CNAs), insertions and deletions (indels), and fusions. DNA and RNA were amplified using the Ion AmpliSeq library kit 2.0 (Thermo Fisher Scientific). To prepare the barcoded library, the Ion Xpress Barcode Adapter 1–96 kit (Thermo Fisher Scientific) was combined with the non-barcoded adapter mix in the Ion AmpliSeq library kit. The resulting amplicons were purified using the Agencourt AMPure XP reagent (Beckman Coulter, Brea, CA, USA) and 70% ethanol, following the manufacturer’s instructions. The concentration of the final library was 50 pM. Libraries were quantified using the Ion Library TaqMan quantitation kit (Thermo Fisher Scientific). The final libraries were transferred to the Ion Chef System (Thermo Fisher Scientific) for automated template preparation. Sequencing was performed on the Ion Torrent S5XL Machine platform (Thermo Fisher Scientific) with an Ion 540 Chip kit (Thermo Fisher Scientific).

### 2.5. Data Analysis

The sequencing data were analyzed using the Ion Torrent software (Torrent Suite 5.10.0 with Ion Reporter 5.2) with a default configuration (Thermo Fisher Scientific) using a medium sensitivity setting for CNA detection. The Ion Torrent software was also used for the alignment of reads to the reference genome (GRCh37-Hg19). Briefly, the criteria of the variant allele frequency for SNVs and indels were ≥4% and ≥7% (hotspot ≥ 3%), respectively. Average CNAs ≥ 4 and <1 were interpreted as a gain (amplification) and loss (deletion), respectively. For translocations, read counts ≥ 20 and total valid mapped reads ≥ 50,000 were interpreted as positive results. The analysis results of most tumor samples were within the standards of sequencing results, such as mapped reads > 5,000,000, an on-target rate > 90%, a mean depth > 1200, and a uniformity > 90%. Results with poor quality and suspected errors were filtered out based on the following criteria: variant allele frequency < 5%; coverage < 100×; variants in the intron region.

### 2.6. Assay Validation

To verify the workflow of the Ion S5XL system, a verification test was performed using commercially available control reference agents of OncoSpan gDNA (Horizon Diagnostics, Cambridge, UK) and the 5-Fusion RNA multiplex positive/negative control (Horizon Diagnostics). NGS reactions were performed using two replicates to validate control testing and two replicates to demonstrate the limit of detection testing.

### 2.7. Statistical Analysis

All statistical analyses were performed using the IBM SPSS statistical software package, version 27 (IBM Corp., Armonk, NY, USA). The significance of the correlation between clinicopathological parameters and *EGFR* status was analyzed using the paired *t*-test, Pearson’s chi-square test, and Fisher’s exact test. Survival data were analyzed using the Kaplan–Meier method and compared using the log-rank test. Differences were considered significant at *p* < 0.05.

## 3. Results

### 3.1. Case Selection and Clinical Parameters

The results of the analysis of 208 tumors from 101 patients with multiple lung cancers diagnosed between 2017 and 2020 were retrieved (Appendix A). The mean age of the patients was 65.89 years (range, 44–82 years; median, 67 years). The number of female patients (*n* = 59; 58.4%) was higher than that of male patients (*n* = 42; 41.6%). Sixty-six patients (65.3%) did not have a smoking history. Multiple lung cancers located in the same lobe were identified in 33 patients (32.7%), whereas bilateral lung tumors were identified in 35 patients (34.7%). The size of the largest tumor was ≤3 cm (<pT2) in 81 patients (80.2%). Lymph node metastasis was identified in 11 patients (10.9%). Nine patients had metachronous multiple lung tumors, and the interval between the tumor appearances was in the range of 7–48 months (mean, 19.6 months). Other demographic characteristics are summarized in Table 1.

### 3.2. EGFR Mutation Status of Tumors

Table 2 and Appendix A summarize the analysis of the *EGFR* mutation status and clinicopathological parameters. The incidence of *EGFR* mutations was high in women (*p* < 0.001) and never-smokers (*p* < 0.001). The duration of smoking was significantly low in patients with *EGFR*-mutant tumors who were ex-smokers or current smokers (*p* < 0.001). Among patients having at least one *EGFR*-mutant tumor, the demographic parameters were not significantly different between patients having all *EGFR*-mutant tumors and those having an *EGFR*-mutant tumor and *EGFR*-wildtype tumor. Additionally, the frequency of *EGFR* mutations was high in cancers occurring in the upper portion of the lung (right upper lobe, right middle lobe, or left upper lobe) (*p* = 0.002).

Comprehensive histopathological and molecular evaluation (Appendix A) revealed that *EGFR* mutations were frequent in adenocarcinomas (*p* < 0.001) with a predominant acinar pattern (*p* < 0.001). In contrast, adenocarcinomas with a predominant lepidic pattern were associated with significantly less frequency of *EGFR* mutations (*p* = 0.017).

### 3.3. Concordance of EGFR Mutation Status

In this study, 52 patients (51.5%) had different *EGFR* mutation statuses (16 patients had different mutations, and 36 patients had at least one *EGFR*-wildtype tumor), 20 patients had the same *EGFR* mutations (19.8%), and 29 patients (28.7%) had only *EGFR*-wildtype tumors. The clinicopathological data and *EGFR* mutation status are schematized in Figure 1. The most frequently detected *EGFR* mutation among the patients with the same *EGFR* mutation was L858R (detected in 15 patients), followed by an exon 19 deletion (detected in 5 patients). The results of comparative analysis of clinicopathological parameters between patients with the same mutation and those with different mutations among patients with *EGFR*-mutant tumors only are summarized in Table 3. Lymph node metastasis was detected in 5 patients with the same *EGFR* mutation, although this was not significant (*p* = 0.053). Other clinicopathological parameters were not significantly different.

As *EGFR*-wildtype NSCLC can exhibit various molecular alterations, all *EGFR*-wildtypes in the same patient may not exhibit the same molecular profile. For an accurate comparison, 24 *EGFR* tumors from 12 patients with *EGFR*-wildtype tumors only, and 9 *EGFR*-wildtype tumors from 9 patients with both *EGFR*-mutant and *EGFR*-wildtype tumors were randomly selected and subjected to NGS analysis. NGS analysis of 33 *EGFR*-wildtype tumors revealed two rare *EGFR* mutations (H773Q in two tumors from one patient and A289V in one tumor from other patient), which could not be detected using the cobas^®^ EGFR mutation test v2. The location, histological diagnosis, and molecular alterations of each tumor for 12 pairs of *EGFR*-wildtype tumors are shown in Figure 2. The results of NGS analysis of tumors from nine patients who had at least one *EGFR*-wildtype tumor are summarized in Appendix A.

We excluded 17 patients with all *EGFR*-wildtype tumors and who were not subjected to NGS analysis. The clinicopathological parameters between molecular-concordant patients and molecular-discordant patients were comparatively analyzed (Appendix A). The discordance rate was 71.4% (60/84). Additionally, the parameters were not significantly different between the molecular-concordant and molecular-discordant patients. Furthermore, the Martini–Melamed criteria [11], ACCP guidelines [12,13], and comprehensive histological assessment proposed by Girard et al. [14] were applied to evaluate the clinicopathological criteria for determining MPLC and IPM. The discriminatory power of these parameters was not significantly different between the concordant and discordant groups.

### 3.4. Survival Analysis

The mean follow-up period was 34.49 months (range, 13.70–76.60 months; median, 30.30 months). During the follow-up period, tumor recurrence was observed in 14 (13.9%) patients, and 4 (4.0%) patients died. The presence of tumors with a size of >3 cm, lymph node metastasis, and smoking history were negatively correlated with disease-free survival (DFS) (Figure 3A–C). The presence of an *EGFR*-mutant tumor was negatively correlated with DFS, although this correlation was not significant (Figure 3D). Sex and the concordance of molecular alterations of multiple cancers did not influence DFS (Figure 3E–F).

## 4. Discussion

In this study, 72 (71.3%) of the 101 patients with multiple lung cancers had *EGFR* mutations in at least one tumor. Consistent with the previous findings, *EGFR* mutations were frequently detected in women (51/72, 69.4%) and never-smokers (58/72, 80.6%) [22,23,24]. Additionally, patients with *EGFR*-mutant/wildtype tumors exhibited similar characteristics to patients with *EGFR*-mutant/mutant tumors when compared with patients with *EGFR*-wildtype/wildtype tumors. The demographic parameters were not significantly different between patients with *EGFR*-mutant/wildtype tumors and those with *EGFR*-mutant/mutant tumors. These findings suggest that patients with multiple lung cancers, at least one *EGFR*, and those with multiple lung cancers with all *EGFR*-wildtype tumors had exhibited different demographic characteristics.

*EGFR* mutation was frequently detected in tumors located in the upper portion of the lung (79/113; 69.9%) [25] and was common in adenocarcinoma (111/113; 98.2%), especially in tumors with an acinar-predominant pattern (85/113; 76.6%) [26]. In particular, *EGFR* mutations were less frequently detected in lepidic-predominant tumors (13/113; 11.7%). Among the lepidic-predominant tumors included in this study, the proportion of less progressed cancers, including MIA, was high. This finding was consistent with the hypothesis that *EGFR* mutations were acquired during the development of invasive adenocarcinoma from pre-invasive lesions, such as AAH, and that they accelerated the progression of tumors rather than initiating tumorigenesis [27]. Additionally, *EGFR* mutations were acquired independently in precancerous lesions with multifocal presentations [9], even though the same patient exhibited different *EGFR* statuses in individual tumors.

This study demonstrated that more than 50% of patients had tumors with different *EGFR* mutation statuses. After excluding patients with *EGFR*-wildtype/wildtype tumors who exhibited demographically diverse characteristics when compared with other patients, the rate of discordance was 72%. The number of patients with both *EGFR*-mutant and *EGFR*-wildtype tumors was 36 (36/72; 50%). Therefore, subjecting only one cancerous lesion to *EGFR* analysis cannot conclusively determine the *EGFR* mutation status in the remaining tumors.

The pathogenic mechanism of multiple lung cancers was explained through the development of metachronous or synchronous tumors in the squamous epithelium of the oral cavity of smokers (‘field cancerization’) [6,28,29]. However, never-smokers account for approximately 25% of lung cancer cases [30]. History of tobacco usage cannot explain the occurrence of cancer in these patients. Previous studies have reported that 70–76% of patients with lung cancer in Korea have a history of smoking [31,32]. In contrast, 65.3% of patients with lung cancer were never-smokers in this study. Recently, the incidence of lung cancer has been increasing in never-smokers [33]. Adenocarcinoma accounts for 40.3–91.3% of all synchronous MPLC cases [5,18,19]. The study cohort comprised a high proportion of Asian female patients with MPLC who were never-smokers [24]. Thus, these findings indicate the importance of screening programs in the never-smoker subgroup [34].

Various studies have examined the pathogenesis and driver mutation of lung cancer in never-smokers. The most common driver mutation of lung cancer in East Asian never-smokers is *EGFR* mutation [24,35,36,37]. Determining the timing of *EGFR* mutations during carcinogenesis can predict whether multiple lung cancers share the same *EGFR* mutation or have different *EGFR* mutations. The development of diagnostic imaging technology has enabled the detection of multiple ground-glass opacities (GGOs) and increased the detection rate of synchronous lung cancer. In particular, TNM classification [38] can distinguish between MPLC and IPM. Therefore, determining if multiple lung cancers develop from MPLC or IPM is critical for accurate diagnosis and the determination of optimal treatment plans.

In the last 35 years, several studies have aimed to accurately diagnose multiple lung cancers [11,12,13,14]. However, differentiating MPLC or IPM based on histological features has been challenging, owing to the heterogeneous histological features of lung cancer. Consistently, differentiating between MPLC and IPM based only on histological features and clinical findings was challenging in this study. The results of this study suggested that the rate of the discordant histological pattern in patients with *EGFR*-mutant/wildtype tumors was higher than that in patients with *EGFR*-mutant/mutant tumors. These findings can be attributed to the differences in histological features between *EGFR*-mutant tumors and *EGFR*-wildtype tumors [24,39]. The concordance of the histological patterns was not significantly different between the molecular-concordant and the molecular-discordant groups. In particular, the discriminatory power of the currently used criteria to distinguish MPLC and IPM was not significantly different between the two groups. Thus, distinguishing tumors based only on clinicopathological findings without molecular findings is a major hurdle. Therefore, a combined histomolecular approach [16] must be utilized for differentiating MPLC and IPM in real-world clinical settings. Additionally, individual tumors of multiple lung cancers must be subjected to molecular analysis.

To evaluate the multiple lung tumors using the combined histomolecular approach, molecular markers other than *EGFR* mutations must be identified. Although the cost effectiveness of NGS is superior compared to the quantity of information provided by NGS, and the turnaround time is reduced due to the development of technology [40,41], the implementation of NGS in all patients with NSCLC, including those with early-stage tumors, is difficult in real-world clinical settings. The administration of adjuvant EGFR-TKI increased DFS in patients with resected *EGFR*-mutant NSCLCs [42,43]. Hence, *EGFR* analysis in all NSCLC lesions is critical for the management of patients with NSCLC in real-world clinical settings. Analysis for other molecular markers or NGS of all NSCLC lesions should be carefully deliberated and validated considering the cost-effectiveness and therapeutic benefits to the patients. Future studies must establish comprehensive histomolecular criteria that can be used as a gold standard in real-world clinical settings.

In patients with multiple lung cancers who underwent surgery for major lesions harboring *EGFR* mutations, postoperative EGFR-TKI treatment for unresected GGO lesions decreased the size of residual GGOs, especially in patients with large or residual GGOs or advanced stage tumors. However, the size of residual GGOs in most patients was unchanged [44]. In our study, only 50% of patients with *EGFR*-mutant tumors had another *EGFR*-mutant tumor. These findings suggest the importance of determining the mutation status of each residual lesion when considering adjuvant EGFR-TKI treatment for patients with multiple lung cancers.

Advances in diagnostic technology have provided various methods to detect *EGFR* mutations, including circulating tumor DNA (ctDNA) analysis (also called liquid biopsy analysis) [45]. Liquid biopsy can confirm the *EGFR* mutation status without the need for a surgical resection of the tumor, which is useful for analyzing *EGFR* mutations in unresectable MPLC cases. However, some studies including our previous study have reported that ctDNA analysis detects *EGFR* mutations in only 54–57% of advanced NSCLC cases with *EGFR* mutations, which was confirmed using tissue DNA analysis [46,47]. Additionally, the sensitivity of ctDNA analysis to detect *EGFR* mutations is low in localized NSCLC [48]. Therefore, the probability of the presence of *EGFR* mutations should be considered even when liquid biopsy analysis does not detect *EGFR* mutations in MPLC. *EGFR* mutation analysis must be performed using all obtainable tumor tissues of MPLC.

This study has some limitations. Only data from patients with multiple lung cancers at the operable stage were examined in this study. Hence, the pathological and clinical stages of lung cancer were relatively low. Consequently, the concordance of *EGFR* mutations in advanced-stage multiple lung cancers has not been verified. Additionally, the follow-up period was not sufficient. In this study, the concordance of *EGFR* mutations did not affect the prognosis; however, the follow-up period was not sufficient to determine whether concordant *EGFR* mutations indicate metastasis or tumors that share the same *EGFR* mutation. These limitations must be addressed in future studies.

## 5. Conclusions

This study revealed the independent occurrence of *EGFR* mutations in patients with multifocal lung cancers. The discordance of *EGFR* mutations in individual tumors was observed in more than half of the patients. Additionally, histological or clinical findings did not successfully predict the presence of concordance. Therefore, the presence of different *EGFR* mutations in multiple lung cancers with the same or different histological characteristics must be considered, except when they are highly predicted to share the same molecular profiles, such as those of IPM.

## Figures and Tables

**Figure 1 cancers-14-03011-f001:**
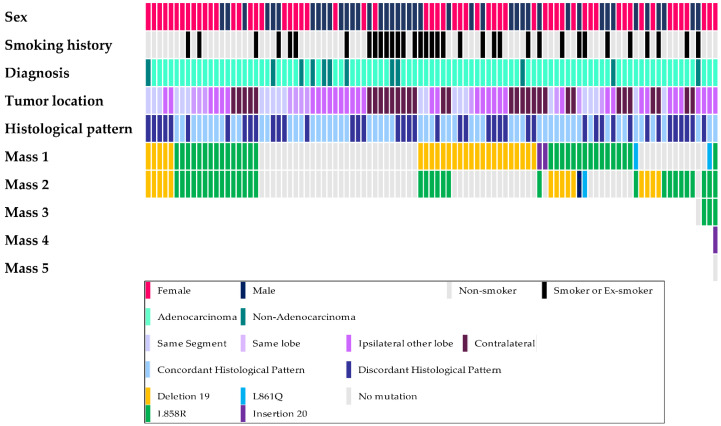
Schematic demonstration of the clinicopathological parameters and *EGFR* mutation.

**Figure 2 cancers-14-03011-f002:**
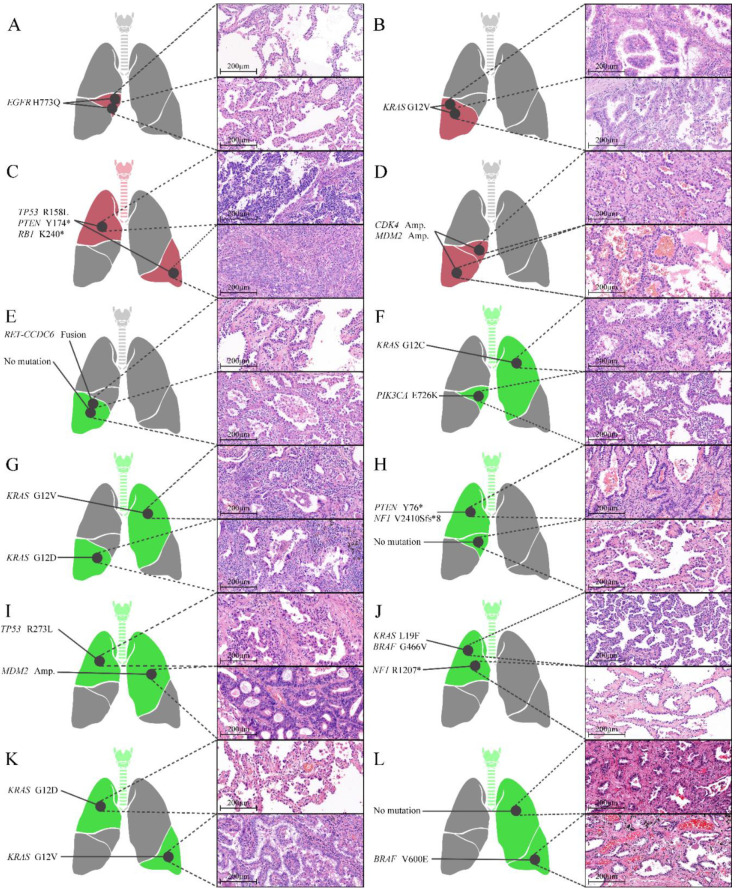
Locations, histological characteristics (H&E-stained sections), and molecular alterations of individual tumors of 12 pairs of *EGFR*-wildtype tumors. Four tumors (**A**–**D**) shared the same molecular alterations (red), while eight tumors (**E**–**L**) exhibited different molecular alterations (green). * stop-gain mutation.

**Figure 3 cancers-14-03011-f003:**
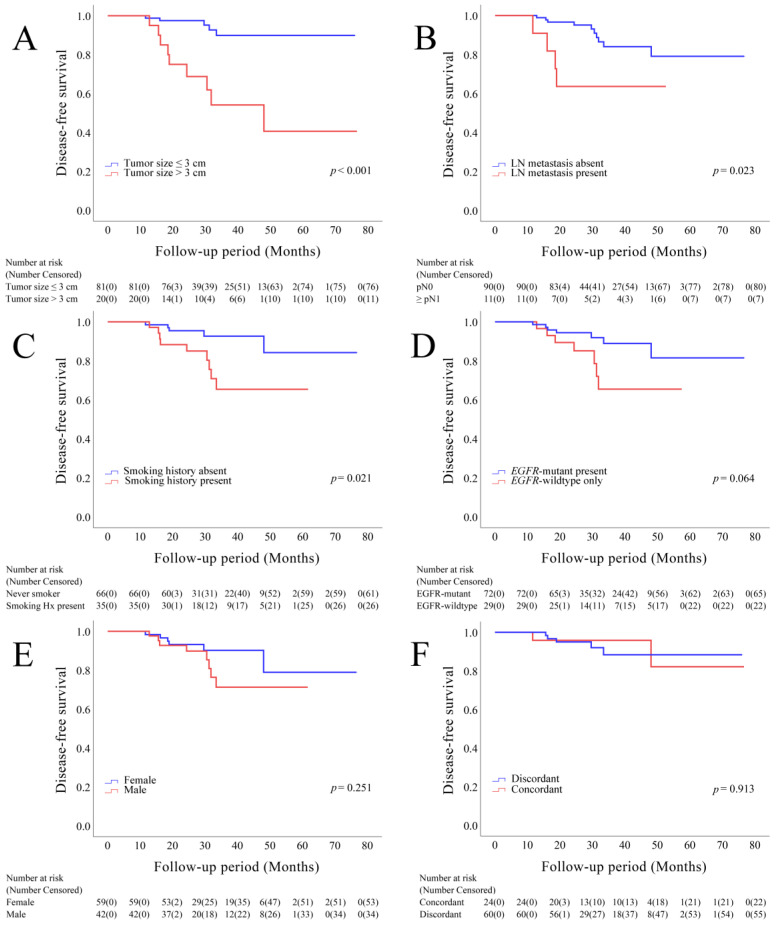
Kaplan–Meier survival curves depicting disease-free survival, stratified based on the size of the largest tumor (**A**), lymph node metastasis (**B**), smoking history (**C**), *EGFR* status (**D**), sex (**E**), and concordance of molecular alterations in individual tumors (**F**).

**Table 1 cancers-14-03011-t001:** Demographic characteristics of patients.

Parameters	Number of Patients (%)
Age (years; mean ± SD)	65.89 ± 8.30
<65	46 (45.6)
≥65	55 (54.5)
Sex	
Male	42 (41.6)
Female	59 (58.4)
Tumor location	
Unilateral side, single lobe	33 (32.7)
Same segment	22 (21.8)
Multiple segments	11 (10.9)
Unilateral side, multiple lobes	33 (32.7)
Bilateral side	35 (34.6)
Number of masses	
2	97 (96.0)
3	3 (2.9)
5	1 (0.1)
Largest tumor size	
≤3 cm	81 (80.2)
>3 cm	20 (19.8)
Lymph node metastasis	
Absent	90 (89.1)
Present	11 (10.9)
Synchronous vs. Metachronous	
Synchronous	92 (91.1)
Metachronous	9 (8.9)
Smoking history	
Absent	66 (65.3)
Present	35 (34.7)
Present smoking status	
Never-smoker	66 (65.3)
Ex-smoker	20 (19.8)
Current smoker	15 (14.9)
Patient status	
Alive	97 (96.0)
Dead	4 (4.0)

Abbreviation: SD, standard deviation

**Table 2 cancers-14-03011-t002:** Comparison of demographic parameters between patients with *EGFR*-wildtype tumors and *EGFR*-mutant tumors.

Parameters	*EGFR*-Mutant (*n* = 72)		*p*-Value	*EGFR*-Wildtype (*n* = 29)	*p*-Value *
	*EGFR*- Mutant/Wildtype (*n* = 36)	*EGFR*- Mutant/Mutant (*n* = 36)			
Sex					
Male	13 (36.1)	9 (25.0)	0.306	20 (69.0)	<0.001
Female	23 (63.9)	27 (75.0)		9 (31.0)	
Age (years; mean ± SD)	64.50 ± 7.54	65.58 ± 8.30	0.564	68.00 ± 9.04	0.130
<65	19 (52.8)	16 (44.4)	0.479	11 (37.9)	0.652
≥65	17 (47.2)	20 (55.6)		18 (62.1)	
Smoking history					
Absent	28 (77.8)	30 (83.3)	0.551	8 (27.6)	<0.001
Present	8 (22.2)	6 (16.7)		21 (72.4)	
Smoking duration (pack–years) ^†^	22.13 ± 12.36	13.00 ± 14.57	0.245	37.81 ± 12.78	<0.001
Present smoking status					
Never-smoker	28 (77.8)	30 (83.3)	-	8 (27.6)	<0.001
Ex-smoker	5 (13.9)	5 (13.9)		10 (34.5)	
Current smoker	3 (8.3)	1 (2.8)		11 (37.9)	

Data are presented as number of patients (%) unless otherwise noted. Abbreviation: SD, standard deviation. * *p*-value represents differences between the *EGFR*-mutant group and *EGFR*-wildtype group. ^†^ Smoking duration was calculated only in patients with a history of smoking.

**Table 3 cancers-14-03011-t003:** Comparison of demographic parameters between patients with same *EGFR* mutation and those with different *EGFR* mutations in the *EGFR*-mutant/mutant group.

Parameter	Same (*n* = 20)	Different (*n* = 16)	*p*-Value
Sex			0.146
Male	3 (15.0)	6 (37.5)	
Female	17 (85.0)	10 (62.5)	
Age (years)			0.940
<65	9 (45.0)	7 (43.8)	
≥65	11 (55.0)	9 (56.3)	
History of smoking			
No	18 (90.0)	12 (75.0)	
Yes	2 (10.0)	4 (25.0)	0.374
Present smoking status			
Never-smoker	18 (90.0)	12 (75.0)	
Ex-smoker	2 (10.0)	3 (18.8)	
Current smoker	0 (0.0)	1 (6.3)	0.175
Largest tumor size			0.455
≤3 cm	16 (80.0)	14 (87.5)	
>3 cm	4 (20.0)	2 (12.5)	
Lymph node metastasis			0.053
Absent	15 (75.0)	16 (100.0)	
Present	5 (25.0)	0 (0.0)	
Tumor location			
Unilateral side, same lobe	9 (45.0)	7 (43.8)	0.762
Same segment	6 (30.0)	5 (31.3)	
Different segment	3 (15.0)	2 (12.5)	
Unilateral side, different lobe	6 (30.0)	3 (18.8)	
Bilateral side	5 (25.0)	6 (37.5)	

Data are presented as number of patients (%).

## Data Availability

All data and materials are available from Y.-L.C. (ylachoi@skku.edu) upon reasonable request.

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
