# Peer review of "The High Proportion of Discordant EGFR Mutations among Multiple Lung Tumors"

_cancers, 2022, doi:10.3390/cancers14123011_

Round 1
Reviewer 1 Report
In this study Lee et al. perform EGFR mutations analysis and an NGS assay on early lung tumors that were surgically resected. Unfortunately, the main message of this paper, that multiple tumors within a given could harbor distinct mutations is well-documented. Moreover, the authors fail to address the most obvious answer to this dilemma in today's clinical practice: liquid biopsy. Liquid biopsy clearly can pick up multiple mutations from different tumors and the authors have not shown liquid biopsy data in this manuscript. Moreover, the authors do not discuss to any significant degree their "proposal for appropriate staging and establishment of suitable treatment."
Author Response
Please see the attachment for detailed information.
In this study Lee et al. perform EGFR mutations analysis and an NGS assay on early lung tumors that were surgically resected. Unfortunately, the main message of this paper, that multiple tumors within a given could harbor distinct mutations is well-documented.
Q1. Moreover, the authors fail to address the most obvious answer to this dilemma in today's clinical practice: liquid biopsy. Liquid biopsy clearly can pick up multiple mutations from different tumors and the authors have not shown liquid biopsy data in this manuscript.
Thank you for your thoughtful comments.
As you mentioned, the liquid biopsy could be good alternative method of EGFR analysis. However, the purpose of our study was evaluation of the prevalence of EGFR mutation and the potential for discordant molecular alterations in each tumors of MPLC. Therefore, we did not collect the result of EGFR analysis using liquid biopsy. Also, according to previous studies, the sensitivity of liquid biopsy was not high enough to detect every EGFR mutation in MPLC, especially in early stage. Therefore, we added a paragraph about liquid biopsy in discussion section (Page 14, Line 412 – Page 15, Line 425).
Q2. Moreover, the authors do not discuss to any significant degree their "proposal for appropriate staging and establishment of suitable treatment".
Thank you for your comments. We deleted the phrase “proposal for appropriate staging and establishment of suitable treatment” from our title.
We appreciate your comments on our manuscript. All comments were very helpful in improving our article, and we did our best to respectfully follow your detailed advice.
The manuscript was revised accordingly, and the changes were highlighted in the manuscript. We hope our work is satisfactory to your comment and suitable for publication.
Once again, thank you for your thoughtful, valuable, and generous comments regarding our study.
Please see the attachment for additional information

Reviewer 2 Report
Authors showed that discordant of driver mutation status is frequently observed in patients with multiple lung cancer. It is considered a useful information for the discussion about treatment strategy including targeted therapy for multiple lung cancer.
Authors reviewed a data of patients who underwent surgery for multiple lung cancer from 2017 to 2020. Of these, patients in whom EGFR mutation was tested are enrolled in this study. Finally, 101 patients showing multiple lesions in whom a data of EGFR mutation was available were assessed.
It is important how many patients were excluded during the patient screening process. Please provide the information on the total number of patients who underwent surgery for lung cancer between 2017 and 2020 and number of patients excluded due to lack of information on EGFR mutation.
Supplementary table 2 shows characteristics of patients with concordant EGFR mutation (n = 49) and discordant EGFR mutation (n = 52).
Authors stated that “after excluding patients with EGFR-wildtype in all tumors that had not been subjected to NGS analysis”. As a result, supplementary table 4 shows a data of patients with concordant EGFR mutation (n = 24) and discordant EGFR mutation (n = 60).
This explanation is confusing for me. Table 1 and 2 showed that 72 patients had at least one tumor harboring EGFR mutation. However, the sum of the number of patients with concordant EGFR mutation (n = 49) and discordant EGFR mutation (n = 52) is 101 in table s2.
Furthermore, EGFR mutation was detected in 72 patients by PCR, and EGFR mutation was subsequently detected by NGS in 9 patients. Thus, a total of 81 patients had lung cancer harboring EGFR mutation. However, table s4 shows there were 84 patients with EGFR mutated lung cancer. And, I wonder why number of patients with concordant EGFR mutation was decreased from 49 in table s2 to 24 in table s4.
Author Response
Please see the attachment for detailed information.
Authors showed that discordant of driver mutation status is frequently observed in patients with multiple lung cancer. It is considered a useful information for the discussion about treatment strategy including targeted therapy for multiple lung cancer.
Authors reviewed a data of patients who underwent surgery for multiple lung cancer from 2017 to 2020. Of these, patients in whom EGFR mutation was tested are enrolled in this study. Finally, 101 patients showing multiple lesions in whom a data of EGFR mutation was available were assessed.
Q1. It is important how many patients were excluded during the patient screening process. Please provide the information on the total number of patients who underwent surgery for lung cancer between 2017 and 2020 and number of patients excluded due to lack of information on EGFR mutation.
Thank you for your precious comment. We performed EGFR analysis on all of cases who underwent surgery for lung cancer. Therefore, there were no patients who did not have information of EGFR mutation. Instead, we provided additional supplementary figure about workflow of patient selections (Figure S4, Page 5).
Q2. Supplementary table 2 shows characteristics of patients with concordant EGFR mutation (n = 49) and discordant EGFR mutation (n = 52). Authors stated that “after excluding patients with EGFR-wildtype in all tumors that had not been subjected to NGS analysis”. As a result, supplementary table 4 shows a data of patients with concordant EGFR mutation (n = 24) and discordant EGFR mutation (n = 60). This explanation is confusing for me. Table 1 and 2 showed that 72 patients had at least one tumor harboring EGFR mutation. However, the sum of the number of patients with concordant EGFR mutation (n = 49) and discordant EGFR mutation (n = 52) is 101 in table s2.
Thank you for your detailed comments, and we apologize for not stating clearly. At first we included EGFR-wildtype/wildtype (N=29) into EGFR-concordant group, therefore the number of EGFR-concordant group was 49 (Same EGFR mutation = 20 / No EGFR mutation = 29).
We decided to delete this confusing result and table from our manuscript, so the original Table S2 was deleted. Instead, we inserted new table (Table 3) about comparison of EGFR-mutant/mutant patients with same mutation and with different mutation. Also, we added additional paragraph about Table 3 in Page 7, Line 240 – 254 and revised manuscript.
Also we revised the term “EGFR-concordant / EGFR-discordant” to “patients with same EGFR mutation / different mutation” or “Molecular-concordant / Molecular-discordant”.
Q3. Furthermore, EGFR mutation was detected in 72 patients by PCR, and EGFR mutation was subsequently detected by NGS in 9 patients. Thus, a total of 81 patients had lung cancer harboring EGFR mutation. However, table s4 shows there were 84 patients with EGFR mutated lung cancer. And, I wonder why number of patients with concordant EGFR mutation was decreased from 49 in table s2 to 24 in table s4
Thank you for your thoughtful comments, and we apologize for not stating clearly. We performed NGS in EGFR-wildtype tumor and there were no EGFR mutation, except one patient with EGFR H773Q, which could not be detected using cobas PCR. We revised manuscript in Page 9, Line 261-273 and supplementary table (Table S3) more clearly.
We appreciate your comments on our manuscript. All comments were very helpful in improving our article, and we did our best to respectfully follow your detailed advice.
The manuscript was revised accordingly, and the changes were highlighted in the manuscript. We hope our work is satisfactory to your comment and suitable for publication.
Once again, thank you for your thoughtful, valuable, and generous comments regarding our study.
Please see the attachment for additional information

Reviewer 3 Report
The clinical data set is valuable. The topic itself, multiple primary versus intrapulmonary metastase or recurrence, is also very valid. However, I struggle to follow the results and the conclusions.
- Title - The title contains the following: ‘Proposal for appropriate staging and establishment of suitable treatment’'. Currently this could not be retrieved from the results, neither has these two points been discussed in the manuscript, therefore this part should be deleted from the title.
- Table 1.
- Is it necessary to define age into four categories, maybe <65 and >65 is sufficient.
- Number of masses - ≥4 – there was one patient with 5 masses, maybe its worth mentioning that: 2-97; 3-3; 5-1
- For metachronous tumours, time interval between tumours should be given
- Smoking status could be presented: never-ex-smoker-current
- Table 1 presents patients’ characteristics, tumour characteristics such as EGFR at least 1 tumour absent/present, does not need to be presented here
- Results
Obviously, if patient has two tumours diagnosed at the same time – these two tumours have had different evolutionary pathways, and EGFR mutation status in one tumour is unrelated to the other.
Following from the above, according to Table 2, 36 patients had one tumour EGFR mutant, one tumour EGFR wild type. For these patients, genomic findings support it is multiple primary.
36 patients who had both tumours EGFR mutant, based on Figure 1 – 15+5 patients had EGFR concordant or same mutations in both tumours. I think it is very difficult to argue that these tumours, particularly if synchronous, are multiple primaries.
For those patients with both tumours EGFR wild type status (29 patients), there is no genomic characteristic.
I struggle to find one common, most important finding in the manuscript. What is the gold standard to define whether it is a new primary or metastases?
For staging, pathological staging is the most accurate and is considered as gold standard, all other methods – CT, PET/CT, EBUS sampling, mediastinoscopy have limited accuracy and are compared to pathological staging (obtained after complete systematic lymph node dissection).
Main histology – small cell, adenocarcinoma, squamous, help discriminate, has much higher predictive power any other clinical criteria. Histology pattern based on morphology is prone to subjectivity.
In line with these thoughts, I suggest that the Authors select one parameter as gold standard, and will compare /validate other findings against it.
None of the existing clinical criteria presents absolute truth, hence, one whether Martin Melamed or ACCP is sufficient. Alternatively, would one perform better than other?
Currently, the data have been presented as EGFR mutation status is gold standard, and this is not true for patients with concordant EGFR alterations.
Table 3 – The Authors compare tumour characteristics in EGFR wild type and EGFR mutant tumours (eg histology, location size). Difference in clinical or tumour characteristics between EGFR mutant and wild type tumours should not be focus of this manuscript.
Rather, the focus could be, could genomic characteristics help us better stratify between multiple primary versus metastases. Figure 2 nicely illustrates that NGS, additional genomic markers provide much better stratification, is very useful in cases where patient’s clinical management would differ. I suggest the Authors focus on genomic markers and on the validity of clinical criteria.
NGS for 143 genomic alterations using Thermo Fisher test was performed. Figure 2 presents data for 12 patients, Supplemental Table 3 for 9 patients. Instead of focusing on EGFR status, the Authors could focus on NGS, present all NGS findings, and discuss whether genomic characteristics help better define multiple primary.
In Abstract: ‘Next generation sequencing (NGS) was performed on tumors with EGFR-wildtype tumors.’
In Supplemental Table 3 - The molecular alterations revealed by next-generation sequencing in patients with EGFR-mutant tumor and EGFR-wildtype tumor.
Clarification is needed how many samples were sequenced.
Line 239 – The Authors state: ‘In our study, 52 patients (51.5%) had different EGFR mutations, and 49 patients (48.5%) had the same EGFR mutations.’
According to Table 2 – (36+36) 72 patients had EGFR mutations, and 29 patients had wild type tumours. Line 245 also confirms 29 patients had no EGFR mutation.
Hence, data in line 239 and in Table 2/line 245 are contradicting.
Figure 1 - Line 242-245 should be part of the main manuscript text, The definition of EGFR concordant should be revised, not EGFR mutant in both tumours, but concordant should indicate the same EGFR mutation in both tumours. Tumours with different EGFR mutation should not be considered concordant (similar to no mutation).
I suggest Table 2 addition EGFR mutant/mutant, should differentiate the same mutation (N=20 based on Figure 2), and 16 pt with different EGFR mutation.
Supplemental Table 2 EGFR concordant and discordant is therefore misleading.
Author Response
Please see the attachment for detailed information.
The clinical data set is valuable. The topic itself, multiple primary versus intrapulmonary metastase or recurrence, is also very valid. However, I struggle to follow the results and the conclusions.
Q1. Title - The title contains the following: ‘Proposal for appropriate staging and establishment of suitable treatment’'. Currently this could not be retrieved from the results, neither has these two points been discussed in the manuscript, therefore this part should be deleted from the title
Thank you for your comments. We deleted the phrase “proposal for appropriate staging and establishment of suitable treatment” from our title.
Q2. Table 1. Is it necessary to define age into four categories? Maybe <65 and >65 is sufficient.
We really appreciate your detailed recommendation. We modified table 1, 2, 3 and S4 as you recommended.
Q3. Number of masses - ≥4 – there was one patient with 5 masses, maybe its worth mentioning that: 2-97; 3-3; 5-1.
Thank you for your thoughtful comments. We revised table 1 as you recommended.
Q4. For metachronous tumours, time interval between tumours should be given.
Thank you for your detailed comments. We added the information of time interval between tumors in metachronous tumors (Page 5, Lines 212-214).
Q5. Smoking status could be presented: never-ex-smoker-current.
Thank you for your thoughtful recommendation. We revised table 1, 2, 3 and S4 as you recommended.
Q6. Table 1 presents patients’ characteristics, tumour characteristics such as EGFR at least 1 tumour absent/present, does not need to be presented here.
Thank you for your detailed recommendation. We revised Table 1 and removed EGFR data from Table 1.
Obviously, if patient has two tumours diagnosed at the same time – these two tumours have had different evolutionary pathways, and EGFR mutation status in one tumour is unrelated to the other. Following from the above, according to Table 2, 36 patients had one tumour EGFR mutant, one tumour EGFR wild type. For these patients, genomic findings support it is multiple primary.
36 patients who had both tumours EGFR mutant, based on Figure 1 – 15+5 patients had EGFR concordant or same mutations in both tumours. I think it is very difficult to argue that these tumours, particularly if synchronous, are multiple primaries. For those patients with both tumours EGFR wild type status (29 patients), there is no genomic characteristic. I struggle to find one common, most important finding in the manuscript. What is the gold standard to define whether it is a new primary or metastases?
For staging, pathological staging is the most accurate and is considered as gold standard, all other methods – CT, PET/CT, EBUS sampling, mediastinoscopy have limited accuracy and are compared to pathological staging (obtained after complete systematic lymph node dissection). Main histology – small cell, adenocarcinoma, squamous, help discriminate, has much higher predictive power any other clinical criteria. Histology pattern based on morphology is prone to subjectivity.
Q7. In line with these thoughts, I suggest that the Authors select one parameter as gold standard, and will compare / validate other findings against it. None of the existing clinical criteria presents absolute truth, hence, one whether Martin Melamed or ACCP is sufficient. Alternatively, would one perform better than other?
We appreciate your thoughtful comments. The purpose of our study was to reveal that the existing recommendations using pathological staging and histology pattern alone were insufficient to discriminate between MPLC and IPM. We wanted to emphasize the importance of comprehensive histomolecular evaluation, and to accentuate that importance of molecular evaluation including EGFR analysis in each tumors of multiple lung tumors. Rather than highlighting only one parameter, we would like to emphasize that both histological and molecular evaluation of multiple lung tumors should be sufficiently performed. So we added some phrases to the discussion in Page 14, Line 387-401. We hope you understand our decisions with wide magnanimity.
Q8. Table 3 – The Authors compare tumour characteristics in EGFR wild type and EGFR mutant tumours (eg histology, location size). Difference in clinical or tumour characteristics between EGFR mutant and wild type tumours should not be focus of this manuscript.
Thank you for your valuable comments and we were totally agree with you. Therefore, we moved table 3 from main manuscript to supplementary materials (New Table S2 (Originally table 3)) (Page 6, Line 217-218)
Q9. Rather, the focus could be, could genomic characteristics help us better stratify between multiple primary versus metastases. Figure 2 nicely illustrates that NGS, additional genomic markers provide much better stratification, is very useful in cases where patient’s clinical management would differ. I suggest the Authors focus on genomic markers and on the validity of clinical criteria.
Thank you for your thoughtful recommendation. We were totally agree with you, however our study was conducted in a real clinical setting. In terms of cost-effectiveness and practicality, NGS is is difficult to fully imply in real clinical setting. At the time of study design we all agreed with that, so we focused on EGFR analysis, which has a great therapeutic significance and the relatively high incidence rate.
We wanted to suggest additional genomic markers based on other genomic markers, but considering purpose and design of our study, it was difficult to apply to our study. We added some sentences in discussion section (Same paragraph we answered in Q7, Page 14, Line 407-422) We hope for your generous acceptance of our decisions.
Q10. In Abstract: ‘Next generation sequencing (NGS) was performed on tumors with EGFR-wildtype tumors.’
In Supplemental Table 3 - The molecular alterations revealed by next-generation sequencing in patients with EGFR-mutant tumor and EGFR-wildtype tumor.
Line 239 – The Authors state: ‘In our study, 52 patients (51.5%) had different EGFR mutations, and 49 patients (48.5%) had the same EGFR mutations.’
According to Table 2 – (36+36) 72 patients had EGFR mutations, and 29 patients had wild type tumours. Line 245 also confirms 29 patients had no EGFR mutation.
Hence, data in line 239 and in Table 2/line 245 are contradicting.
Thank you for your thoughtful comments, and we apologize for not stating clearly. We performed NGS in EGFR-wildtype tumor and there were no EGFR mutation, except one patient with EGFR H773Q, which could not be detected using cobas PCR. We revised manuscript in Page 9, Line 261-273 and supplementary table (Table S3) more clearly.
Q11. Figure 1 - Line 242-245 should be part of the main manuscript text, The definition of EGFR concordant should be revised, not EGFR mutant in both tumours, but concordant should indicate the same EGFR mutation in both tumours. Tumours with different EGFR mutation should not be considered concordant (similar to no mutation). I suggest Table 2 addition EGFR mutant/mutant, should differentiate the same mutation (N=20 based on Figure 2), and 16 pt with different EGFR mutation. Supplemental Table 2 EGFR concordant and discordant is therefore misleading.
Thank you for your detailed comments, and we apologize for not stating clearly. At first we included EGFR-wildtype/wildtype (N=29) into EGFR-concordant group, therefore the number of EGFR-concordant group was 49 (Same EGFR mutation = 20 / No EGFR mutation = 29).
We decided to delete this confusing result and table from our manuscript, so the original Table S2 was deleted. Instead, we inserted new table (Table 3) about comparison of EGFR-mutant/mutant patients with same mutation and with different mutation. Also, we added additional paragraph about Table 3 in Page 7, Line 240-254 and revised manuscript.
Also we revised the term “EGFR-concordant / EGFR-discordant” to “patients with same EGFR mutation / different mutation” or “Molecular-concordant / Molecular-discordant”.
We appreciate your comments on our manuscript. All comments were very helpful in improving our article, and we did our best to respectfully follow your detailed advice.
The manuscript was revised accordingly, and the changes were highlighted in the manuscript. We hope our work is satisfactory to your comment and suitable for publication.
Once again, thank you for your thoughtful, valuable, and generous comments regarding our study.

Reviewer 4 Report
1. The introduction section is too short and needs to be reinforced with more scientific data as it mostly contains generalized sentences. Please add scientific data, hypothesis and author’s point of view why the study is necessary and what specific benefit these types of studies might bring to patients with multiple tumors.
2. Please briefly comment in the discussion section on the high percentage of never-smokers in your group of patients with multiple cancers, and the need for the implementation of screening programs in this patient subgroup as well (example reference: Kerpel-Fronius et al. JTO, 2022. https://www.jto.org/article/S1556-0864(21)02391-1/fulltext)
3. Considering the sentence in the discussion: “These findings suggest that it is crucial to determine the 383 mutation status of each residual lesion when considering adjuvant 384 EGFR-TKI treatment for patients with multiple lung cancer.” Please briefly continue to comment on the possibility to achieve this in everyday clinical practice. Please put the obtained results in the context of applicability in a diagnostic/clinical center in a real-world setting.
4. As the topic is up to date, please update your reference list accordingly. The current reference list includes too many references dating from before 2010. Please replace/add more references from 2020, 2021, 2022.
5. The Manuscript would benefit from a polishing by a native speaker – the sentences can be well understood but the reading flow would be better if some expressions were re-formulated.
Author Response
Please see the attachment for detailed information.
Q1. The introduction section is too short and needs to be reinforced with more scientific data as it mostly contains generalized sentences. Please add scientific data, hypothesis and author’s point of view why the study is necessary and what specific benefit these types of studies might bring to patients with multiple tumors.
Thank you for your detailed comments. We added paragraphs in Page 2, Line 94 - Page 3, Line 106 in introduction section.
Q2. Please briefly comment in the discussion section on the high percentage of never-smokers in your group of patients with multiple cancers, and the need for the implementation of screening programs in this patient subgroup as well (example reference: Kerpel-Fronius et al. JTO, 2022. https://www.jto.org/article/S1556-0864(21)02391-1/fulltext)
Thank you for your thoughtful recommendation. We added paragraph in Page 13, Lines 346-354 in discussion section.
Q3. Considering the sentence in the discussion: “These findings suggest that it is crucial to determine the 383 mutation status of each residual lesion when considering adjuvant 384 EGFR-TKI treatment for patients with multiple lung cancer.” Please briefly continue to comment on the possibility to achieve this in everyday clinical practice. Please put the obtained results in the context of applicability in a diagnostic/clinical center in a real-world setting.
Thank you for your valuable recommendation. We added paragraphs in Page 14, Line 387-401 in discussion section.
Q4. As the topic is up to date, please update your reference list accordingly. The current reference list includes too many references dating from before 2010. Please replace/add more references from 2020, 2021, 2022.
Thank you for your comments. However, our old references was inserted for definition. Therefore, we added some up-to-date references rather than exclusion of old references. We hope for your generous acceptance of our decisions.
Q5. The Manuscript would benefit from a polishing by a native speaker – the sentences can be well understood but the reading flow would be better if some expressions were re-formulated.
Thank you for your valuable recommendation and we all agree with your comments. We sent our manuscript to a native speaker for polishment of our manuscript.
We appreciate your comments on our manuscript. All comments were very helpful in improving our article, and we did our best to respectfully follow your detailed advice.
The manuscript was revised accordingly, and the changes were highlighted in the manuscript. We hope our work is satisfactory to your comment and suitable for publication.
Once again, thank you for your thoughtful, valuable, and generous comments regarding our study.

Round 2
Reviewer 1 Report
The authors have made sufficient modifications.
Author Response
We appreciate the in-depth review of our manuscript entitled “High proportion of discordant EGFR mutations among multiple lung tumors”. We did our best to correct minor methodological errors and spell errors.
The manuscript was revised accordingly, and the changes were highlighted in the manuscript. We hope our work is satisfactory to the reviewer’s comment and suitable for publication in the Cancers. Looking forward to hearing good news from the editorial and the reviewers.

Reviewer 4 Report
Thank you for the detailed review.
Author Response

(The authors gave the same response as above.)
